# An Analysis of Mechanisms for Cellular Uptake of miRNAs to Enhance Drug Delivery and Efficacy in Cancer Chemoresistance

**DOI:** 10.3390/ncrna7020027

**Published:** 2021-04-16

**Authors:** Justine M. Grixti, Duncan Ayers, Philip J. R. Day

**Affiliations:** 1Department of Biochemistry and Systems Biology, Institute of Systems, Molecular and Integrative Biology, Biosciences Building, University of Liverpool, Liverpool L69 7ZB, UK; justine.grixti@liverpool.ac.uk; 2Centre for Molecular Medicine and Biobanking, University of Malta, Msida MSD 2080, Malta; 3Faculty of Biology, Medicine and Human Sciences, The University of Manchester, Manchester M1 7DN, UK; Philip.J.Day@manchester.ac.uk

**Keywords:** miRNA, Drug Delivery, cancer chemoresistance, drug resistance, cell uptake

## Abstract

Up until recently, it was believed that pharmaceutical drugs and their metabolites enter into the cell to gain access to their targets via simple diffusion across the hydrophobic lipid cellular membrane, at a rate which is based on their lipophilicity. An increasing amount of evidence indicates that the phospholipid bilayer-mediated drug diffusion is in fact negligible, and that drugs pass through cell membranes via proteinaceous membrane transporters or carriers which are normally used for the transportation of nutrients and intermediate metabolites. Drugs can be targeted to specific cells and tissues which express the relevant transporters, leading to the design of safe and efficacious treatments. Furthermore, transporter expression levels can be manipulated, systematically and in a high-throughput manner, allowing for considerable progress in determining which transporters are used by specific drugs. The ever-expanding field of miRNA therapeutics is not without its challenges, with the most notable one being the safe and effective delivery of the miRNA mimic/antagonist safely to the target cell cytoplasm for attaining the desired clinical outcome, particularly in miRNA-based cancer therapeutics, due to the poor efficiency of neo-vascular systems revolting around the tumour site, brought about by tumour-induced angiogenesis. This acquisition of resistance to several types of anticancer drugs can be as a result of an upregulation of efflux transporters expression, which eject drugs from cells, hence lowering drug efficacy, resulting in multidrug resistance. In this article, the latest available data on human microRNAs has been reviewed, together with the most recently described mechanisms for miRNA uptake in cells, for future therapeutic enhancements against cancer chemoresistance.

## 1. Transport Mechanisms: Introduction to Membrane Transporters

### 1.1. Role of Membrane Transporters in Cellular Uptake

Up until recently, it was believed, and in some text books it is still portrayed, that pharmaceutical drugs, and their metabolites, enter into the cell to gain access to their targets, via simple diffusion across the hydrophobic lipid cellular membrane, at a rate which is based on their lipophilicity [1,2,3,4]. An increasing amount of evidence indicates that the phospholipid bilayer-mediated drug diffusion is in fact negligible [1,5], and that drugs pass through cell membranes via proteinaceous membrane transporters or carriers which are normally used for the transportation of nutrients and intermediate metabolites, as shown in Figure 1 below [2,6,7,8,9,10,11,12,13,14].

Recognition that the flux of metabolites and pharmaceutical drugs into biological cells occurs through membrane transporters has important and beneficial implications. Drugs can be targeted to specific cells and tissues which express the relevant transporters, leading to the design of safe and efficacious treatments. Furthermore, transporter expression levels can be manipulated, systematically and in a high-throughput manner, allowing for considerable progress in determining which transporters are used by specific drugs [15].

### 1.2. Membrane Transporters

A membrane transporter can be defined as an integral membrane protein [16] which regulates the transportation of physiological nucleosides and nucleobases for nucleic acid synthesis in the salvage pathways, metabolites and exogenous dietary products, as well as being the gatekeepers of transport of many drugs across the cell membrane [16,17,18,19,20,21,22]. The role of the eukaryotic cell membrane thus provides physical confines for all types of differentiated cells to conduct vital physiological processes, allowing only selected metabolites and/or key molecules to enter the cell cytoplasm, and similarly permits the efflux of specified endo-metabolites, ions, synthesised products, and ‘waste’ or toxic compounds into the extra cellular environment for systemic dispersion and/or removal.

Transporters comprise solute carriers, ion channels, water channels, and ATP-driven pumps [23]. Uniporters are integral membrane proteins that allow facilitated diffusion, binding to one molecule of substrate at a time, transporting it down its concentration gradient. Typical examples of uniporters include those associated with amino acid or glucose movement across the cell membrane. On the other hand, symporters and antiporters are involved in secondary active transport, where ATP, generated through an electrochemical gradient, is used to transport molecules against a concentration gradient. Symporters transport molecules in the same direction with respect to each other, in contrast to the antiporter transporter. Movement of ions across the plasma membrane is down their concentration gradient. This facilitated diffusion is coupled with active transport, allowing for the other molecule(s) to be transported against their concentration gradient. Moreover, primary active transporters, transport substrates against their concentration gradient, deriving energy directly from the breakdown of ATP (see Figure 1).

In the context of cancer chemoresistance and miRNA influences on such tumour characteristics, the tumour cell membrane can play essential roles in allowing access to regulatory miRNAs present (either in free circulation or bound within exosomes) in the bloodstream. Considering that previous literature on this research niche is still in its infancy, the review describes possible manners in which the cell membrane-based compound uptake mechanisms act, for future clinical exploitation in developing more effective drug delivery of miRNA (or other non-coding RNA)-based therapeutics. However, for readers who are interested in identifying miRNAs that directly affect membrane transporter- and/or drug efflux transporter-related genes, the comprehensive review by Ayers and Vandesompele (2017) on this matter is freely available within the scientific literature, focusing solely on miRNAs and lncRNAs involved in cancer chemoresistance [24].

## 2. Transport Mechanisms across the Cell Membrane

Transport through transporters can be both passive and active. ‘Active’ transport occurs when the flux in or out of the cell at the expense of an energy source. ‘Passive’ transport is when not coupled by an energy source expenditure and simply flows from a region of high-concentration to a region of low-concentration. However, it is imperative to distinguish that passive transport does not mean that transportation through the cellular bilayer occurs via diffusion and is not carrier-mediated [4]. When a process is not energy coupled, thermodynamics terms this as passive in nature. However, this has nothing to do with its molecular mechanism.

As a result, transporters in humans are categorised into two major superfamilies, based on their thermodynamic properties and ATP coupling, namely the solute carrier (SLC) superfamily, which are generally influx transporters, and the ATP-binding cassette (ABC) superfamily, which are typically efflux transporters [11,21,22,25,26,27].

## 3. Chemoresistance

Primary treatment for both early and advanced tumours is achieved through chemotherapy. Unfortunately, drug resistance seriously limits the efficacy of conventional chemotherapeutics and novel biological agents, hence resulting in a major obstacle in the treatment of cancer. Drug resistance can be intrinsic in cases where the tumour is insensitive to therapeutic agents prior to any treatment. Otherwise, resistance is defined as acquired if the tumour develops resistance during the course of treatment [28]. This acquisition of resistance to several types of anticancer drugs can be as a result of an upregulation of efflux transporters expression, which eject drugs from cells, hence lowering drug efficacy, resulting in multidrug resistance [14,29]. Other mechanisms which contribute to drug resistance include insensitivity to apoptosis induced by drugs, increased repair of DNA damage, and recent data demonstrates that drug resistance might also be conferred to ncRNAs [28,30]. The next sections of this review article are, in fact, dedicated to the different transport mechanisms of ncRNAs, mainly miRNAs, and how these link to chemoresistant tumours.

## 4. Non-Coding RNAs

The term non-coding RNA (ncRNA) is commonly used to refer to RNA which does not encode a protein. However, this by no means implies that such RNA molecules serve zero functions. Recent advances in technology resulted in the revolutionisation of the molecular world and have shown that the majority of mammalian and genomes of other complex organisms is, in fact, transcribed into ncRNAs. Such RNA molecules form a hidden layer of internal signals which in turn control various levels of gene expression in physiology and development, including chromatin structure, epigenetic memory, transcription, RNA splicing, editing, translation, and turnover [31]. The miRNA family of non-coding RNAs is individually composed of an RNA duplex approximately 22 base pairs in length, with one strand being complementary to specific target transcripts and ultimately leading to post-transcriptional repression of such transcripts [32,33,34,35,36]. The specific number of ncRNAs within the human genome is unknown. These are classified on the basis of their size; transcripts shorter than 200 nucleotides, including miRNAs, siRNAs, piRNAs, are referred to as ncRNAs. Transcripts with a length between 200 nt and 100 kb make up the second group, known as lncRNAs [31]. miRNAs is the class of ncRNAs which is most frequently studied. miRNAs are estimated to regulate ~30% of all protein-coding genes and are fundamental in shaping the global transcriptome of eukaryotes [37,38]. A more comprehensive overview of all ncRNA families can be found though the open access review by Micallef and Baron, published just recently [39].

### MicroRNA Functions and Their Clinical Importance

Presently, there are over 2000 human miRNAs that have been identified, validated and catalogued following global efforts in novel miRNA discovery [40,41,42].

At the genomic levels, miRNAs are located in intronic areas of host genes or within intergenic regions [43]. Additionally, further research highlighted the fact that over 50% of all miRNAs expressed in humans fall within cancer-associated genomic regions or within fragile sites [44].

The initial stages of miRNA processing begin with transcription of the miRNA-bearing host gene within the cell nucleus into a pri-miRNA by the enzyme RNA polymerase II (Pol II) [45]. The pri-miRNA is essentially a long primary transcript, which is consequently targeted for cleavage by the nuclear enzyme Drosha into a precursor RNA (pre-miRNA), a stem-loop structured RNA approximately 60–70 base pairs in length [46]. The pre-miRNA is then transported out of the nucleus and into the cell cytoplasm by means of an Exportin-5/Ran-GTP enzyme complex [46]. Once the pre-miRNA is released into the cytoplasm, the RNAse III enzyme Dicer binds onto its target pre-miRNA, resulting in the formation of a mature miRNA duplex (miRNA/miRNA*) of approximately 22 nucleotides in length but having two 3′ overhangs at both ends [47,48]. Finally, the mature miRNA duplex is then bound by the RNA-induced silencing complex (RISC), whereby the passenger mRNA* strand of the miRNA duplex is cleaved [49]. Consequently, the remaining miRNA strand of the mature miRNA acts as a guide for its bound RISC enzyme, with this strand being either totally or partially complementary to a unique sequence on the target mRNA [49].

Two possible modes of action exist by which the miRNA/RISC complex manages to inhibit translation of the target mRNA. If there is perfect complementarity between the miRNA and its target mRNA sequence, the latter is cleaved by means of the endonucleolytic properties of the Argonaute 2 (AGO2) domain present in RISC [50]. Alternatively, the miRNA binds to the 3′ untranslated region (UTR) of the target mRNA with imperfect complementarity on the 3′UTR seed region (nucleotides #2-8), thus repressing translation of the target mRNA into its intended protein products [50].

The involvement of miRNA activity in shaping the development and severity of a vast spectrum of human disease conditions, including breast, lung, gastric, and liver cancers, HIV, influenza virus, multiple sclerosis, and type II diabetes [51], has undoubtedly affirmed their clinical importance [52,53,54,55,56,57,58,59,60,61,62,63,64,65,66,67,68,69,70,71]. This specific involvement via dysregulated miRNAs can enable their exploitation as reliable biomarkers for specific disease presence and/or clinical progress [72,73,74,75,76,77,78,79,80,81]. In addition, miRNA biomarkers can also qualify as novel drugs or drug targets for the potential development of novel miRNA-based therapeutics [82,83,84,85,86,87,88,89,90,91]. The mechanisms involved in enabling the physical uptake of circulating miRNAs was long thought to be based an ATP-independent diffusion through the individual cell membrane. However, emerging evidence in scientific literature relates the existence of exosomes that have the capacity to transport miRNAs through the systemic circulation to distal tissues.

## 5. Drug Delivery Issues in miRNA Therapeutics

The ever-expanding field of miRNA therapeutics is not without its challenges, with the most notable one being the safe and effective delivery of the miRNA mimic/antagonist safely to the target cell cytoplasm for attaining the desired clinical outcome, particularly in miRNA-based cancer therapeutics, due to the poor efficiency of neo-vascular systems revolting around the tumour site, brought about by tumour-induced angiogenesis [92]. This challenge presents itself since the chemical composition of the vast majority of such novel therapeutic drugs is RNA-based, and therefore their direct administration in the patient bloodstream can lead to nuclease-directed elimination of the drug, the activation of the innate immune system mechanisms and/or immunotoxicity and neurotoxicity (particularly for miRNA mimic-based drugs) and also due to the sheer dose of the drug required for effective pharmacodynamic profiles [92,93,94]. Finally, even though the therapeutic miRNA payload can successfully reach the target tissues, there still remains the challenge of attaining sufficient intracellular delivery of the miRNAs (normally through endosomal formation) by effective endosomal escape mechanisms [92].

### 5.1. Current miRNA Drug Delivery Methods

#### 5.1.1. Local Administration of miRNAs

The prospect of locally administering miRNA-based therapies directly into a solid tumour mass or immunoprivileged sites carries significant advantages, most notably the enhanced pharmacodynamic profile due to evasion of nuclease activity on the individual miRNA- oligonucleotide drug, together with reduced toxicity effects, since lower doses can effectively be administered for implementing the pharmacological effects required [92]. Several studies have successfully introduced miRNAs in a direct or topical manner for multiple conditions, including glioblastoma [95,96,97,98].

Such a method of administration can be highly effective for all conditions in which the site of pharmacological action can be accessed physically, including early stage cancer patients having a defined tumour mass. However, for other medical conditions and late stage/metastatic cancer contexts, local administration of miRNA-based therapies might not be the ideal therapeutic option.

#### 5.1.2. Systemic Administration of miRNAs

The main advantage of administration of miRNA-based therapies in a systemic manner is to allow for the pharmacological action to affect ‘hard to reach’ target tissues afflicted by clinical conditions having miRNA dysregulated expression hallmarks. However, as described above, there are numerous challenges for such therapies to exert their intended functions efficiently and safely. Consequently, many bespoke miRNA-based drug delivery methods are currently in development to ensure safe passage of the oligonucleotides through the bloodstream and eventual target tissue, ultimately leading to cellular uptake of the oligonucleotides. Described briefly below are the current miRNA delivery methods being developed globally (see Table 1).

##### Chemical Modifications on miRNA Structure

(a)2′OH group modification

The ribose ring-based 2′ OH group is particularly exposed to nuclease activity and therefore is a challenge for the employment of chemically unmodified miRNA antagonists and mimics as a systemic-administered therapy [99]. Consequently, modification of the 2′ OH group (typically through 2′-*O*-methylation) is a highly effective means for preventing such modified oligonucleotides from nuclease degradation [99,100,101].

(b)Locked nucleic acids (LNAs)

Locked nucleic acids represent a conformational modification of the RNA backbone that can interact more avidly with the intended complimentary miRNA target sequence for which the miRNA antagonist/mimic has been designed [136]. This modification method is best represented through the successful development by Santaris [Denmark] of a LNA-antimiR for miR-122, capable of forming stable heteroduplexes with the targeted miRNA sequence [102]. The study also demonstrated that the effect of the LNA-antimiR for miR-122 was effective in reducing hepatitis C infection in vivo, in a dose dependent manner and no (murine) hepatotoxicity [102]. This novel technology has also been expanded to develop tiny LNAs with the capacity to bind avidly to 8-mer seed sequences common to entire families of miRNAs, such as the oncogenic miR-17~92 and miR-106b~25 clusters and have already demonstrated success in murine medulloblastoma studies [137,138,139].

(c)Passenger strand alterations and carrier vehicles

Modifications of the passenger strand apply only to miRNA mimic-based therapeutics, since there exists the presence of the RNA duplex. Such modifications can aid the degree of protection for the miRNA mimic from nuclease activity while in the bloodstream and also lead to reduced immunotoxicity effects, mainly due to Toll-Like Receptor-induced interferon systemic release [103]. In addition, such modifications do not affect the functionality and efficacy of the guide strand of the miRNA mimic [140,141].

Notwithstanding the efficacy of miRNA modifications in allowing increased success rates for the miRNA antagonist/mimic delivered safely to the site of pharmacological action, the efficiency in actual drug uptake by the target tissue component cells could still be enhanced through the use of carrier vehicles, such as biodegradable, biocompatible and non-toxic biopolymers including chitosan, cyclodextrins, poly-l-lysine, dextran, poly (lactic co-glycolic acid), polyglutamic acid, hyaluronic acid and gelatin [142].These drug delivery systems have the added advantage (apart from protecting the drug during bloodstream passage) of having higher bonding affinity and interactions with the target tissue cell membrane, therefore enhancing the degree of drug uptake by the target cells [143].

##### Viral-Based Delivery Systems

The employment of viruses for drug payload protection and effective uptake by target tissues has long been evaluated for multiple drugs bearing a precariously fragile nature. This is in particular so for biologics-based drugs, as broadly described since the beginning of the gene therapy era in the early 1990s [144,145,146,147]. In the context of miRNA therapeutics, viral delivery systems are most apt for the transport of vectors coding for specific miRNA mimic/antagonist sequences, which also allows for a longer duration of action of the miRNA therapy within the target cells [104,105,106,107,108,109,110,111,112,113,114,115,116,117,118,119,120].

However, issues with risk of insertional mutagenesis of viral genomic sequences within patient tissues, ease of preparation and scalability of the viral delivery systems presently hinder such a drug delivery technology from entering the clinical setting on a widespread basis [148,149].

##### Non-Viral-Based Delivery Systems

(a)Polymer nanoparticles

The utilisation of polymer-based nanoparticles for enabling the delivery of a spectrum of drugs, including miRNAs, is becoming evermore commonplace with versatile and non-immunogenic polymers such as polyethylenimine and polyethylene glycol being utilised as nanoparticle backbones [121,122,123,124,125]. Typical advantages of the use of polymers for nanoparticle drug delivery systems include a highly flexible drug release kinetic profile on response to acid exposure [93].

(b)Inorganic nanoparticles

The use of non-polymer and non-immunogenic materials for the development of nanoparticles capable of drug delivery (including miRNAs) has also demonstrated considerable success.

One of the mainstay inorganic compounds employed for such drug delivery purposes is silica [126,127,128,129,130,131]. The study carried out by Stallings and colleagues was successful in utilising silica nanoparticles for the delivery of miR-34a in neuroblastoma murine tumour xenograft models [129].

The study performed by Yu and colleagues illustrated the efficiency of the utilisation other suitable inorganic materials, such as biodegradable bioactive glass nanoparticles, for the successful delivery of miRNAs [150]. The results of this study also highlighted the improved loading capacity of such glass nanoparticles compared to other inorganic compound-based nanoparticles such as mesoporous silica nanoparticles [150].

(c)Lipid-based vehicles

Undoubtedly, the efforts carried out by Mirna therapeutics [TX, USA] earlier in this decade have led to clinical trials for the first ever miRNA replacement therapy for lung cancer treatment [132]. The study conducted by Wiggins and colleagues demonstrated that miR-34a was downregulation in non-small-cell lung cancer (NSCLC) patients and reverting the expression level through artificially induced upregulation inhibited multiple NSCLC cell line growth [132]. The study also successfully demonstrated the effect of miR-34a upregulation within in vivo tumour xenograft models for NSCLC, utilising a neutral, lipid emulsion-based delivery vehicle [132].

This miRNA delivery method was also employed in the study conducted by Wu and colleagues, studying the possible use of miR-29b as a separate miRNA replacement therapy for NSCLC [133]. The variation employed for this specific miRNA delivery mechanism was namely the use of cationic lipoplexes rather than neutral lipid emulsions [133]. The utilisation of such a cationic lipid (1,2-di-*O*-octadecenyl-3-trimethylammonium propane—DOTMA, as chloride salt) enhances the degree of interplay between the negatively charged target cell membrane and the DOTMA-induced positively charged lipocomplex surface, resulting in higher transfection efficacy and uniformity of miRNA uptake by the target cells [133]. Cholesterol was also utilised for the miRNA delivery vehicle due to its protective role against the miRNA mimic oligonucleotide degradation and since it also aids interaction between the target cell membrane and the miRNA-bearing lipoplexes, resulting in enhanced miRNA uptake by the target cells [133]. It is also noteworthy to mention the use of lipid nanoparticle-based measures can be of utility in delivering miRNA-based therapeutics in the near future [151,152,153].

(d)Folate–miRNA conjugates

Interestingly, the study carried out by Orellana and colleagues in 2017 adopted the use of folamiRs—namely, the direct attachment of miRNAs to folate which enhances the possibility of such miRNA conjugates to be taken up by (tumour) cells overexpressing folate receptors [134]. This study successfully introduced FolamiR-34a (folate–miR-34a conjugate) by triple-negative breast cancer cells at both in vitro and in vivo levels [134].

##### Exosomes

Recent evidence has demonstrated the existence of exosomes that are essentially minute membrane vesicles carrying a myriad of intracellular compounds and proteins [154]. Such exosomes can also be secreted into the systemic circulation for eventual downstream uptake by cells from distal tissues, allowing for remote effector function from the exosomal host cells on release of the key compounds and/or proteins released upon exosome uptake by distal cells [155,156,157]. The actual uptake mechanism of such exosomes can either occur through receptor-mediated endocytosis, with consequential release of the hydrophilic exosomal contents, or through direct fusion of the exosome with the target cell membrane, releasing contents directly into the cytoplasm of the target cell [158,159,160].

The effector functions of exosomes can also be applied for gene regulatory purposes due to the efficient transfer of miRNAs withing exosomes. Zheng and colleagues very recently identified the key roles played by exosomal transfer of tumour-associated macrophage-derived miR-21 for the development of cisplatin resistance by gastric carcinoma cells [135]. In addition, multiple evidence within the scientific literature exists that describes the effects of exosomal transfer of miRNAs that induce tumourigenicity and other key phenotypic characteristics of tumours, particularly between the bone marrow adipose and multiple myeloma cell populations [161].

Although still in its infancy, the existence of exosomal transfer of miRNAs for tumour development can be exploited by artificial development of exosomes carrying miRNA antagonists/mimics for deployment and cellular uptake within the same distal tissues.

## 6. Conclusions and Perspectives

The clinical importance of optimally functioning molecular transport mechanisms within the cell, regardless of the tissue type, can never be underestimated. This statement in favour of the PDIN hypothesis is reflected by the myriad of medical conditions, a handful of which are described below, that arise or are aggravated due to improper/lack of effectiveness in channelling specific molecular players across the cell membrane of the afflicted tissue structures.

The utility of copper as an essential molecular component for enabling effective cellular-level physiological functions, such as angiogenesis, wound healing and shielding from reactive oxidative stress, essentially depends on the degree of presence of human copper transporter 1 (hCTR1) bound onto the cell membrane [162,163,164]. Consequently, altered hCTR1 protein expression levels can be detrimental to the cell physiology. In addition, the hCTR1 can also aid the uptake of platinum through the cell membrane, which has been demonstrated to affect the platinum chemoresistance levels of muscle-invasive bladder cancer patients due to low uptake of the metal when hCTR1 cell membrane presence is low [165].

The transport of cholesterol across the cell membrane is handled by the transmembrane efflux pump known as Niemann–Pick disease type 1 (NPC1) [166]. Functional issues relating to NPC1 ultimately lead to the accumulation of cholesterol, together with cytotoxic agents such as daunorubicin, in the affected cells’ endosomal/lysosomal infrastructure within the cell cytoplasm [166]. Eventually, such efflux pump dysfunction directly contributes to the development of Niemann–Pick disease and the emergence of cancer chemoresistance properties by the affected tissues [166]. One recent study has found miR-33 to have regulatory roles on NPC1 [167].

It is hoped that further evolution of such technologies, all pertaining to the study and analyses of drug uptake mechanisms, could lead us to further evidence for this matter. Eventually, this gain in knowledge can be employed for the development of more effective drugs having much improved pharmacodynamic profiles and consequently allowing such novel drugs to achieve clinically therapeutic levels at much lower doses in the near future through maximised efficiency of cellular uptake mechanisms.

## Figures and Tables

**Figure 1 ncrna-07-00027-f001:**
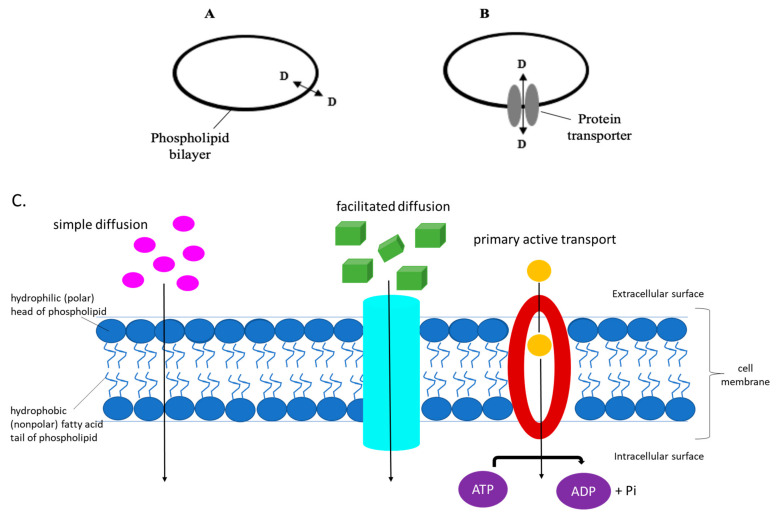
Diagrams of the means by which a molecule, such as a drug, is transported across the cell membrane. (**A**) Diffusion through the phospholipid bilayer, at a rate which is dependent on the substrate’s lipophilicity, and (**B**) via membrane protein transporters which determine the drug’s flux and hence distribution and efficacy. (**C**) The eukaryotic cell membrane and its components, depicting the different ways of how substrates are transported intracellularly.

**Table 1 ncrna-07-00027-t001:** Representation of differing modalities for systemic administration of miRNAs.

Delivery Method	miRNA/s Involved	Reference/s
*miRNA chemical modifications*		
2’ OH- addition on ribose-ring	Multiple miRNAs	[99,100,101]
Locked Nucleic Acids	miR-122	[102]
Passenger Strand Alterations	Multiple miRNAs	[103]
*Viral-based delivery*	Multiple miRNAs	[104,105,106,107,108,109,110,111,112,113,114,115,116,117,118,119,120]
*Non-viral-based delivery*		
Polymer Nanoparticles (e.g., PEG; polyethylenimine)	Multiple miRNAs	[121,122,123,124,125]
Inorganic Nanoparticles (e.g., Silica)	miR-34a	[126,127,128,129,130,131]
Lipid-based Vehicles	miR-34a; miR-29b	[132,133]
Folate—miRNA Conjugates	miR-34a	[134]
*Exosomes*	miR-21	[135]

## Data Availability

Not applicable.

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
