# Peer review of "An Analysis of Mechanisms for Cellular Uptake of miRNAs to Enhance Drug Delivery and Efficacy in Cancer Chemoresistance"

_ncrna, 2021, doi:10.3390/ncrna7020027_

Round 1

Reviewer 1 Report

The review manuscript by Grixti et al.  focused on chemoresistance and miRNA therapeutics. Overall, the review lacks a clear and organized structure. This is reflected in a general feeling of disconnect between sections. Below are my main observations:

  • The first instance of a disconnect is section 1. Are the authors trying to connect the role of miRNAs in chemoresistance by de-regulation of membrane transporters? Or, are the authors trying to argue that miRNAs as chemo-therapeutics don’t face the same obstacles as traditional drugs? This is not clear from the beginning of the text. If the focus is to discuss how manipulating miRNA levels could lead to reducing chemoresistance, the authors need to provide examples of miRNAs that have been linked to chemoresistance by targeting membrane transporters (in the case of an antagomiR approach), or examples of miRNA downregulation that leads to increase levels of efflux transporters (in the case of a miRNA mimic approach).
  • In section 4, the authors should discuss miRNA biogenesis, this is particularly important to divide the two possible options in miRNA replacement therapies 1) viral (using biogenesis pathway) or a synthetic mimic (that does not need pre-processing).
  • Section 5 is the main section that needs a better structure. For instance, section 5.1.2.1 mixes chemical modifications on miRNA mimics and chemical modifications that are used in antagomiRs. These should be treated independently because the therapeutic goal is the opposite and the mechanism of action is different between the two approaches. Also, it is not clear why these modifications are required?
  • Section 5.1.2.1 c, is supposed to discuss passenger strand alterations (which would be stabilizing mods like 2OMe), but in the text discusses carrier vehicles. This is puzzling.
  • Section 5.1.2.3 Discusses non-viral delivery approaches. This section needs to be updated to include receptor mediated delivery of miRNAs (an example is Orellana et al., 2017. Science Translational Medicine), an approach that has been used successfully in siRNA based drugs and has a lot of potential.
  • Section 6 seem completely out of place.
  • Section 7 is completely disconnected to miRNA. The authors discuss about hCTR1, NPC1 is there a miRNA that targets these transporters?

Minor:

 Various typos in the text (i.e. sentences 42, 91, 106)

Reviewer 2 Report

The authors have reviewed mechanisms by which chemotherapeutic drugs and subsequently miRNAs enter the cells. Given the multidrug resistance in Cancers, a review of cellular drug uptake warrants publication. However, major changes are needed to be considered for publication.

Minor Changes:

Typography and Grammar:  with respect to typographical and grammatical errors- few mistakes have crept in that must be checked diligently; few examples are: Line 14. Negligable (needs to be corrected).  Line 42: error with reference- but that sentence structure doesn't read well. The manuscript has additional typographical and grammatical errors that the authors must diligently check and change.

Figures: 1, 2 must be combined to produce 1 image of quality that meets standards of the journal. ,Both figures are basic and just referencing to reviews or additional material to read about them should cater to a larger audience. Appropriate references must be cited if a single figure cannot be produced. Maybe wise to include efflux pumps into this and demonstrate active and passive transport in the same figure.

Fig 3: has extreme similarity with the cited reference- it doesn't add much to the information than what can be gleaned from the cited reference and therefore can be excluded- given that the authors of this review and the cited reference are from the same university- it may be best to reproduce the original figure with permission from the authors, should there be a compelling need to use this figure.

Major Changes

  1. Abstract: miRNA information is too basic and hence may not be needed. May be better to rewrite the abstract to indicate the reason as to why a compilation of methods by which miRNA uptake and efficacy of delivery is necessary.
  2. Role of membranes in cellular uptake is centred on pharmaceutical drugs; the authors need to rework how miRNA fits into this paradigm. Additionally, the role of membrane transporters in pumping drugs out of the cells is important to write about. Often the chemo-resistance that occurs is due to the efflux pumps that is ATP-dependant and has broad range of specificity. The authors mention it but do not provide any details for this. there is a wealth of literature that speaks about multidrug resistance. specific examples for drugs and pumps can be provided to make this review a useful one.  Information about the efflux pumps and multi drug resistance is strewn in several areas- this needs to be better structured. This must be included; specifically relevant information about PgP/ MDR1 must be discussed.
  3. Active and passive transport are described in a very complicated manner, besides "concentrative" in its meaning represents concentration by removing water (or solvent) and is just wrong in this instance. These needs to be described in a simple manner. For e.g. “across a membrane from a region of lower to higher…. At the expense of ATP”. reference appropriate articles when possible
  4. Lines 91-92 is incomplete. The authors need to proof-read and ensure submission of the final version of the manuscript. Also see comments for fig 3.
  5. ncRNAs: the topic is widely discussed and reviewed before. This write up is too basic, it needs to be rewritten to substantiate and introduce the upcoming section.
  6. 341- no need to abbreviate Copper if its not going to be used again
  7. Section 5 needs to be expanded to bring the focus of the review to this topic. with a title like " Analysis of mechanisms of cellular uptake of miRNAs...." it is imperative that this be the champion of the paper. this section needs to be massively expanded to include information  about currently ongoing trials that either use miRNA or the methods such as lipid nanoparticles and the efficacy of therapy in these instances

The positives: Section 5 is better written, this is a warranted area of review

Reviewer 3 Report

The review presented here is well structured, referenced and easy reading. In my opinion, the different delivery methods for ncRNAs are just too lightly described, but if the unique objective of this manuscript is to offer a first contact to this theme, I recommend it for publication.

Minor changes.

Line 42: Error has to be solved.

Line 77: Words/lines missing.

Line 91: Words/lines missing.

Line 107: Correct "treatment".

Line 116: Corrent "chemorresistant".

Reviewer 4 Report

Authors have done a good job in the ideation and presenting its many aspects in the field of cancer research. In the manuscript authors have attempted to present their focus on miRNA-based therapeutics, which is in stages of infancy. Often, miRNAs have been encouraged as potential biomarkers for several complex human diseases including cancer, neuroinflammation and cardiovascular diseases. Herein, the information discussed is inadequate and lacks details raising a conflict with the title of the manuscript. The concept of “chemo-resistance” needs to be clarified better with examples and should be made cohesive in the context of the story.

Major Gap1- The manuscript does not elaborate on the mechanisms underlying transport/shuttling of nucleic acids, instead a few textbook level images (figures 1-2) are used to present the general idea which is unfair given the ambitious/challenging nature of this concept. Tentative Reference- https://www.sciencedirect.com/topics/biochemistry-genetics-and-molecular-biology/nucleic-acid-transport#:~:text=The%20nucleic%20acid%20cargoes%20from,acids%20such%20as%20small%20RNAs.

Major Gap2- Authors must elaborate and discuss on the ongoing efforts in the field of nucleic acid therapeutics, first. Then can segue into miRNA-based, given their specific interests. Interestingly, in the manuscript there is no evidence/ updated information on the most promising miRNA-based strategies or therapeutics that may be in clinical or pre-clinical trials in the field of cancer and human diseases.

Major Gap 3- Lack of clear and detailed illustrations, and well-planned tables (for instance, name of industry/pharma; target; strategy; disease; clinical trial status) highlighting the most promising nucleic acid/miRNA-based therapeutics will be extremely helpful for the audience.

Additional important comments-

  1. Authors have failed to leverage on the promising mRNA based COVID-19 therapeutic efforts by Moderna and Pfizer. Details of their respective technologies can be found in these links- https://www.fda.gov/media/144434/download;

https://www.fda.gov/media/144246/download

  1. How does the pharmacokinetics of miRNAs differ from small molecules, for instance, explaining this concept is of high value for the field and audience since vast majority of therapeutics for cancer and other diseases are based on “small molecule” technology?

  2. What are the different types of payloads and new therapeutic modalities that can be used to improve miRNA-based therapeutics? for instance, siRNA-GalNac conjugation, triantennary GalNac etc.) that can be used for miRNA delivery as a drug? Must be explained with informative and clear illustrations for mechanism of action.
  3. Classification of challenges for intracellular and targeted deliveries for miRNAs or, nucleic acid-based. Do secretory RNAs interfere with therapeutic miRNAs or RNAs?
  4. Lack of relevant examples supporting emerging ideas- for example, authors haven’t mentioned anything on the first FDA approved small (si)RNA-based therapeutic, Onpattro (Patisiran) from Alnylam, USA. The technologies used here in should be highlighted showcasing the scope of the field.

Relevant bibliography for authors to refer and consider-

https://onlinelibrary.wiley.com/doi/pdf/10.1111/tra.12606

https://www.ncbi.nlm.nih.gov/pmc/articles/PMC6085463/

https://www.sciencedirect.com/science/article/pii/S0168365919305796

https://www.nature.com/articles/s41422-020-0389-3

-----

https://www.frontiersin.org/articles/10.3389/fgene.2019.00478/full

https://www.ncbi.nlm.nih.gov/pmc/articles/PMC6599191/

https://www.cell.com/molecular-therapy-family/nucleic-acids/fulltext/S2162-2531(17)30190-7

https://pubmed.ncbi.nlm.nih.gov/31936122/

https://pubmed.ncbi.nlm.nih.gov/31324016/

https://www.ncbi.nlm.nih.gov/pmc/articles/PMC7226753/

https://www.frontiersin.org/articles/10.3389/fphar.2018.01113/full

https://pharmrev.aspetjournals.org/content/72/3/639

https://pubmed.ncbi.nlm.nih.gov/16159017/

https://www.sciencedirect.com/science/article/pii/S016372581830216X

https://www.tandfonline.com/doi/full/10.1080/15476286.2019.1593094

https://www.nature.com/articles/s41419-020-2571-4

-----

https://www.clinicaltrialsregister.eu/ctr-search/search?query=Cancer+and+microRNA

https://clinicaltrials.gov/ct2/show/NCT01231386

Round 2

Reviewer 1 Report

The authors have addressed most of this reviewer's comments. One thing that is still missing is to connect the targets the authors highlight in the perspective section to miRNAs. The authors should consider including the known information regarding miRNA-mediated posttranscriptional regulation of the two transporters mentioned in the text. In the authors' response they mention that "The section describes areas for research expansion, in order to identify such miRNAs affecting the above-mentioned genes as possible novel miRNA-based therapies."; however, this is not mentioned in the section.

Reviewer 2 Report

The authors have addressed my concerns adequately. I have no additional concerns at this time. The review meets the standards to be accepted for ncRNA.

Author Response

Thank you

Reviewer 4 Report

Authors have done a very good job of upgrading their manuscript. The new version is substantially improved and harbors scientific-relevant information, to the topic discussed/presented. The section 5.1.2 adds a lot of importance on the chemical-based modifications to improve therapeutic index of miRNAs. One suggestion to the authors here will be to incorporate some schematic/representation of high quality showing these chemical modifications and examples of miRNAs harboring any of these modifications as examples in the figure/schematic. "A picture speaks more than 1000 words, as they say". So, please note that a good pictorial representation will be extremely significant and helpful to the audience and the field. Importantly, a well presented review with ideas will add value as well as gain tremendous visibility and citations.

Thank you!
